# Starch-Rich Microalgae as an Active Ingredient in Beer Brewing

**DOI:** 10.3390/foods11101449

**Published:** 2022-05-17

**Authors:** Giorgia Carnovale, Shaun Leivers, Filipa Rosa, Hans-Ragnar Norli, Edvard Hortemo, Trude Wicklund, Svein Jarle Horn, Kari Skjånes

**Affiliations:** 1Division of Biotechnology and Plant Health, Norwegian Institute of Bioeconomy Research (NIBIO), NO-1431 Ås, Norway; giorgia.carnovale@nibio.no (G.C.); filipa.rosa@nibio.no (F.R.); hansragnar.norli@nibio.no (H.-R.N.); 2Faculty of Chemistry, Biotechnology and Food Science, Norwegian University of Life Sciences (NMBU), NO-1432 Ås, Norway; shaun.allan.leivers@nmbu.no (S.L.); trude.wicklund@nmbu.no (T.W.); svein.horn@nmbu.no (S.J.H.); 3Nøgne Ø—Det Kompromissløse Bryggeri A/S, Lunde 8, NO-4885 Grimstad, Norway; edvard.hortemo@nogne-o.no

**Keywords:** *Tetraselmis*, microalgae, brewing, food, beverage, starch

## Abstract

Microalgal biomass is widely studied for its possible application in food and human nutrition due to its multiple potential health benefits, and to address raising sustainability concerns. An interesting field whereby to further explore the application of microalgae is that of beer brewing, due to the capacity of some species to accumulate large amounts of starch under specific growth conditions. The marine species *Tetraselmis chui* is a well-known starch producer, and was selected in this study for the production of biomass to be explored as an active ingredient in beer brewing. Cultivation was performed under nitrogen deprivation in 250 L tubular photobioreactors, producing a biomass containing 50% starch. The properties of high-starch microalgal biomass in a traditional mashing process were then assessed to identify critical steps and challenges, test the efficiency of fermentable sugar release, and develop a protocol for small-scale brewing trials. Finally, *T. chui* was successfully integrated at a small scale into the brewing process as an active ingredient, producing microalgae-enriched beer containing up to 20% algal biomass. The addition of microalgae had a noticeable effect on the beer properties, resulting in a product with distinct sensory properties. Regulation of pH proved to be a key parameter in the process.

## 1. Introduction

Microalgae cultivation is one of the emerging technologies in food sciences because of the rich nutritional profiles, impressive health benefits, and positive environmental impact of these photosynthetic organisms [1,2]. Amongst the wide diversity of microalgal species, *Tetraselmis chui* is particularly interesting for industrial applications in food as its upscaled cultivation is already well established for the production of feed ingredients [3]. Biomass from *Tetraselmis* species was proved to be notably rich in antioxidant content and activity [4,5,6], and species from this genus are reported to have high productivity rates [7]. Additionally, *T. chui* has been proved to be safe for human consumption [8], and has recently been authorised as a novel food and as a food supplement by the European Union (EU 2017/2470 Regulation). Species from the genus *Tetraselmis* have been successfully introduced as a novel ingredient in a wide variety of applications, such as bread [9,10], soup [11], gluten-free bread [12], and savoury biscuits [13]. Products enriched with up to 4% *T. chui* show enhanced bioactive properties and good technical properties, proving the feasibility of microalgal biomass use as an ingredient in staple foods [9,14]. 

With the rapid development of the brewing industry, and of its environmental impact, microalgae are being widely studied for CO_2_ capture and wastewater bioremediation [15], mainly with the focus on removing excessive nutrients and converting the biomass to biofuels [16]. However, microalgal biomass is rarely used as an ingredient in the brewing process. 

The cyanobacteria *Spirulina* has reportedly been used on multiple occasions as a food colouring for seasonal beers, as, for example, in the case of the “Gimmicky Green” beer produced by the Captain Lawrence Brewery in New York. Furthermore, the Dutch microbrewery De Koperen Kat has been crafting an “Algenbier”, adding 5% *Chlorella vulgaris* biomass specifically cultured for this purpose [17]. Despite the reported success of these products, no scientific data are available to assess the role of algal addition on the technical, sensory, and/or bioactive properties of beer brewing. Nonetheless, the bioactive properties of a functional alcoholic beverage, prepared extracting *C. vulgaris* biomass with the Brazilian spirit cachaça, has been tested on mice. The microalgae-enriched beverage was reported to have neuroprotective and antioxidant activity when compared to control cachaça, possibly mitigating the effect of alcohol on neural networks [18]. 

Microalgal biomass has the potential to be introduced as an active ingredient in brewing since microalgae are known to be highly efficient starch producers [19]. Under environmental stress conditions, the carbon partitioning in microalgal cells is redirected towards starch metabolism. *Tetraselmis* species have been widely studied for their potential to produce high starch content under nitrogen deprivation, reaching up to 60% of the total dry weight [20], and *T. chui* was recently reported to produce up to 59% starch under nitrogen deprivation [21].

The aim of this study was to test the feasibility of starch-rich microalgal biomass used as an active ingredient in fermented beverages. The effect of different algae-to-malt ratios on brewing protocols and on the final product was assessed by monitoring the release of fermentable sugars during mashing and fermentation phases, and finally, qualifying the beer characteristics and fermentation efficiencies. Providing insights and methods to implement microalgal starch in brewing may pave the way for an increased microalgal presence in industrial setups and in everyday products. 

## 2. Materials and Methods

### 2.1. Biomass Production

#### 2.1.1. Microalgal Cultivation

*Tetraselmis chui* SAG 8−6 from the SAG Culture Collection of Algae (Göttingen, Germany) was kept on L1 medium agar plates [22] at 22 °C with 20 µmol m^−2^ s^−1^ irradiation. The algae were scaled up, maintaining exponential growth, to 0.1–0.25–0.5–1 L Erlenmeyer flasks in 2× F medium [23] at 50–100 µmol m^−2^ s^−1^ irradiance, on a shaking plate, and supplied with 1% CO_2_: air mixture diffused through a needle. The contents of the Erlenmeyer flasks were then used to inoculate first a GemTube RD1–25, 25 L tubular photobioreactor (LGem BV, De Kwakel, The Netherlands) at 25 ± 2 °C, pH 7.8 ± 0.2 (controlled by CO_2_ addition), and 100 µmol m^−2^ s^−1^ irradiance with 2× F medium. The 25 L culture was then finally scaled up to a GemTube RD1−250, 250 L tubular photobioreactor (LGem BV, De Kwakel, The Netherlands) at 25 ± 2 °C, pH 7.8 ± 0.2 (controlled by CO_2_ addition), and 100 µmol m^−2^ s^−1^ irradiance. 

The cultures were monitored daily by measuring absorbance at 750 nm and nitrogen content with nitrate strips 10–500 mg L^−1^ (NO_3_^−^), MQuant^®^ (Supelco, Bellefonte, PA, USA).

Once the nitrate dropped below 50 mg L^−1^, samples were extracted daily for starch analysis. 

#### 2.1.2. Microalgal Biomass Harvesting and Milling

The culture was harvested after a total 8 days of growth, when starch content higher than 40% had been measured. The algae biomass was centrifuged at 3000× *g* with an Evodos 10 centrifuge (Evodos BV, Raamsdonksveer, The Netherlands). The concentrated algal paste (~15–20% dry weight) was frozen at −20 °C and further freeze-dried in a FreeZone freeze dryer (Labconco, Kansas City, MO, USA). Subsequently, the algae powder was milled in a Planetary Ball Mill PM400 (Retsch GmbH, Haan, Germany) at 400× *g* rpm, for three cycles of 6 min duration (3 + 3 in each direction). Between cycles, the containers were cooled on ice to prevent biomass overheating by friction. Milled biomass was mixed and aliquoted in 2 mL Eppendorf tubes according to protocols for compositional analyses and experiments.

#### 2.1.3. Microalgal Biomass Analysis

Starch content was assessed using the Megazyme Total Starch (AA/AMG) assay kit (Megazyme, Wicklow, Ireland) on 10 ± 1 mg biomass, following the protocol supplied by the manufacturer, adapted to remove chlorophyll interference, as previously reported [24]. Protein content was assessed on 10 ± 1 mg biomass with the Bio-Rad protein assay dye reagent, following the protocol provided by the manufacturer (Bio-Rad, Hercules, CA, USA). Bovine serum albumin was used to create a standard curve. Additionally, 30 ± 1 mg biomass was assessed for fatty acid profile using gas chromatography–mass spectrometry (GC-MS). The fatty acids were identified and quantified as fatty acid methyl esters (FAME), against FAME-mix 37, CRM47885 (Sigma-Aldrich, Laramie, WY, USA). All biomass analysis was performed on quadruplicate samples.

### 2.2. Mashing Experiments with ALGAL Biomass

#### 2.2.1. Microalgal Starch Degradation by Barley Enzymes

The first trials aimed at understanding whether the tightly packed, small, microalgal starch granules would be efficiently degraded by barley starch-degrading enzymes at mashing conditions.

Nine samples of milled starch-rich algal biomass, and nine samples of barley malt standard (K-MALTA kit, Megazyme, Wicklow, Ireland), as a control, were prepared, weighing 10 ± 0.2 mg in 2 mL screw-cap tubes. Enzyme-rich wort was prepared separately with a solid:liquid ratio of 0.24 kg L^−1^, thus adding 24 g of milled Pilsner malt (Weyermann, Bamberg, Germany) in 100 mL water. The flask was incubated for 1 h in a water bath set at 67 °C, shaking the bottle every 5 min. After incubation, the wort was centrifuged at 4700× *g* rpm for 5 min, and 1 mL of the enzyme-rich supernatant was pipetted into each of the pre-weighed microalgae and barley malt standard samples. The tubes were then incubated for 0, 1, and 2 h at 67 °C. At each time point, triplicates of each treatment were inactivated at 95 °C for 5 min, then centrifuged at 15,000× *g* rpm for 5 min. The supernatant was removed, and the pellets were analysed following the protocol described in Section 2.1.3.

#### 2.2.2. Mashing with Malt and Microalgae

After studying microalgal starch degradation by malt enzymes, mashing trials were carried out with microalgae (starch-rich *T. chui)*, barley malt, (Pilsner malt, Weyermann, Bamberg, Germany), and mixtures of microalgae plus barley malt, as shown in Figure 1. The aim of these experiments was to evaluate the behaviour and the effect on fermentable sugar yields of microalgal biomass alone (M1–3), and of 20%, 12.5%, and 5% microalgae substitution of total solids during mashing, compared to barley malt alone (B1–4). One-step mashing at 67 °C was thus carried out in triplicate, in falcon tubes. Figure 1 reports the exact grams of biomass that were used in a 15 mL volume for each trial, where the control had a final solid:liquid ratio of 0.24 kg L^−1^. Microalgae addition in the mash increased the pH to 8 ± 0.2, thus microalgae plus barley malt mixtures were tested with and without pH adjustment to pH 5.5 ± 0.1 with lactic acid, sampling at 0, 1, 2, 3, and 5 h from incubation. 

At each time point, all triplicates were sampled by aliquoting 1 mL into an Eppendorf tube and incubating at 95 °C in a heating block for 5 min to inactivate all enzymes. The samples were then centrifuged at 15,000× *g* rpm for 2 min and the supernatant was stored, at a 1:1 dilution, in 10 mM sulphuric acid (H_2_SO_4_) for further HPLC quantification of maltose and glucose.

### 2.3. Brewing Experiment

Brewing with three malt:algae ratios was tested at a small scale against a 100% malt control. Each treatment was performed in triplicate, and the 12 beers were prepared over 2 consecutive days. Table 1 reports the ingredients for 0.8 L volume, including the hops added at the beginning (60 min) and end of boiling (0 min). Both Pilsner malt and Cara malt were sourced from Weyermann (Bamberg, Germany).

For each batch, malt and algae biomasses were weighed (Table 1) and added to 800 mL water, adjusting pH with lactic acid to the starting value of 5.7 ± 0.1 in 1 L borosilicate glass bottles. The bottles were placed in a water bath set at 67 °C, and temperature increase inside the bottles was measured with an immersion temperature probe. The bottles were incubated for 1 h after the internal temperature of the mash reached 66 ± 2 °C (45 min from start). Samples for high-performance liquid chromatography (HPLC) were extracted at the start, at temperature stabilization, and then after 30 min and 1 h incubation at 66 ± 2 °C. Each mash was then transferred into a 2 L Erlenmeyer flask, filtering out the solids with a colander of 2 mm pore size. Proper lautering and sparging were not easily achievable at such a small scale; thus, the final wort contained a significant amount of sediment. The wort in 2 L Erlenmeyer flasks was set to boil on heating plates for 1 h, Northern Brewer hop (CraftCo, Sofiemyr, Norway) was added at the beginning, and Saaz and Cascade (CraftCo, Sofiemyr, Norway) at the end, of the boiling period. The wort was then adjusted, diluting it with water, to reach a concentration of 12 ± 1 °Brix, measured with a Brix refractometer, and were finally transferred into clean 1 L glass bottles after sampling for further HPLC analysis. After the brew had cooled to 21 ± 1 °C, *Saccharomyces cerevisiae* commercial strain US-05 (Fermentis, Lesaffre, France) was added, according to manufacturer’s instructions, and the bottles were closed with yeast locks.

The brews were incubated at 20 ± 1 °C in the dark and sampled daily via brix refractometry to evaluate progression of the fermentation (results not shown). Fermentation in both batches stopped after 5 days; however, the batches were bottled on the same date, on day 5 for Beers 3 and 4, and on day 6 for Beers 1 and 2. Upon bottling, samples for HPLC analysis were extracted, and the brews were gently decanted into 500, 330, and 250 mL dark glass bottles, removing most of the sediment. Carbonating sugar (sucrose 5 g L^−1^) was added to the beers before capping, and the bottles were incubated for 12 days at room temperature, and for 3 weeks at 4 °C, for maturation.

The beer was finally tasted by a small group of professional brewers from Nøgne Ø brewery, and brewing research scientists from NMBU and NIBIO that provided some general remarks concerning the taste profile. 

### 2.4. Ethanol and Apparent Degree of Fermentation (ADF)

After maturation, the beers were characterized using a Packaged Beverage Analyzer for Beer (PBA-B) instrument (Anton Paar, Graz, Austria), consisting of a DMA 4500 M density meter, an Alcolyzer Beer ME module, a CarboQC ME module, and a PFD filling device. The instrument was operated with the aid of the Generation M software v2.42 (Anton Paar, Graz, Austria).

### 2.5. HPLC Analysis of Glucose and Maltose

All samples were initially diluted to 1:1 with 10 mM H_2_SO_4_ and stored at 4 °C before further analysis. In some instances where sediments occurred in storage over time, samples were centrifuged at 14,800× *g* rpm for 2 min prior to analysis. Sugar concentrations were determined by HPLC using an Agilent Ultimate 3000 (Agilent Technologies, Santa Clara, CA, USA) coupled to an ERC RefractoMax 520 (Shodex, München, Germany) refractive index (RI) detector. An organic acid resin column (Rezex ROA-Organic Acid H^+^ (8%), 300 × 7.8 mm, Phenomenex Inc., Torrance, CA, USA) was used for separation at a temperature of 65 °C and a flow rate of 0.6 mL min^−1^. As the mobile phase, 5 mM H_2_SO_4_ was used. Separation was performed in isocratic mode.

### 2.6. Data Analysis

Data were analysed using R Studio version 3.6.1 (R. RStudio, Inc., Boston, MA, USA). One-way ANOVA was performed on the data after assessing homogeneity of variances with a Bartlett test. Where outliers were detected, a square root transformation was applied. If the transformation did not remove the effect of the outlier, the significance level to reject the null hypothesis was set at *p* < 0.001.

## 3. Results and Discussion

### 3.1. Microalgal Biomass Production and Characteristics

Figure 2 shows dry weight and starch accumulation during the cultivation of *T. chui* in a 250 L photobioreactor. After five days of cultivation, most of the nitrogen content was consumed (NO_3_ < 50 mg L^−1^), and the monitoring of starch content in the biomass began. After three days of starvation, starch content had quadrupled, reaching 50% of the total dry weight. This is in agreement with previously reported data, which indicate that *Tetraselmis* species under nitrogen deprivation can accumulate starch up to 62% of the total dry weight [20,21]. The culture was harvested and freeze-dried, yielding a total 370 g of dry algal powder. The analysis of biomass composition, reported in Table 2, shows that the biomass had a relatively low protein content of 20%; *Tetraselmis* species grown under optimal conditions can reach up to 30% protein [25,26]. The lower level of protein in our study can be explained by the fact that the exhaustion of nitrogen reduces cell functions and protein synthesis, but also by the dilution effect of starch accumulation on the relative content of other components in the cells. The total fatty acid content in the microalgal biomass was approximately 5.2%; this is in agreement with previous studies on *Tetraselmis marina*, which have shown a reduction in fatty acid content under nutrient stress [27]. Polyunsaturated fatty acids, known for their antioxidant properties, represent 52% of the total content. The oxidation of lipids during brewing may result in undesirable flavours, thus it is often important to reduce their content [28]. However, novel research is shedding light on the role of fatty acids in malt biomass, which is approximately 1–3% of the total, proving they influence the properties and qualities of the grains in brewing [29].

### 3.2. Mashing Trials

Firstly, we tested whether the starch granules produced by the microalgae were efficiently digested by barley starch-degrading enzymes activated during the mashing phase (Figure 3). At the incubation temperature of 67 °C, our results show effective digestion of both the algae starch and the malt control after one hour of incubation, when 88 and 95% of the total initial starch content in the *T. chui* and barley malt, respectively, had been degraded. After two hours incubation, the starch in both treatments had nearly been depleted. 

Mashing experiments were then performed on barley malt, microalgae, and microalgae–barley mixtures, with and without pH adjustments. In the 100% malt controls, the sugar release was directly proportional to the amount of malt used in the mashing, for both glucose and maltose (Figure 4a,b).

To test the presence and activity of endogenous algal starch-degrading enzymes, a second mashing trial was performed with only starch-rich algal biomass, incubated in suspension at 67 °C for five hours. Although under nitrogen deprivation endogenous starch degrading enzymes may be synthesized in several microalgal species [30], in this study, either they were not present in significant amounts, or they were deactivated by temperature, since no significant increase in fermentable sugars was detected with HPLC (Figure 4c,d). The measured sugar concentrations were low and close to the detection limit of the HPLC, explaining the relatively high standard deviation. The three concentrations of algae tested correspond to the final contents to be added in mashing and brewing trials with algae–malt mixtures (20, 12.5, and 5% of the total solids).

Measurement of pH upon mashing with barley malt and algae mixtures highlighted a strong effect of algal biomass, resulting in pH values up to 8.0 ± 0.2. In traditional mashing, the pH is usually between 5.2–5.6; in our experiments, samples containing barley malt only (B1–4) fall within this range. Compared to the barley malt mashing (Figure 4a,b), the addition of algal biomass to barley malt without the adjustment of pH (Figure 4e,f) resulted in an overall lower yield of fermentable sugars. This is likely to be a consequence of the increase in pH. The highest yields of glucose and, especially, maltose, were achieved in the samples containing the lowest algal biomass. Consequently, an additional mashing experiment was performed on barley malt and microalgae mixtures, adjusting the pH to a value of 5.5 ± 0.1 with lactic acid at the start of incubation; this yielded excellent results with regards to the production of fermentable sugars (Figure 4g,h). The addition of algal biomass in these samples resulted in an overall reduction in the differences between the three ratios, compared to the former trials, suggesting that microalgal starch is contributing to the pool of total fermentable sugars. Additionally, the final sugar concentrations of both glucose and maltose were closer to the 100% malt control. Samples with the lowest algae content were, however, still the ones achieving the highest sugar concentration; this may be due to the fact that the barley malt used in this trial contains approximately 60% starch, whereas the algae biomass contains 50%. 

### 3.3. Brewing Trials

The results from the former trials were used to scale the experiment to 1 L bottles, adjusting the pH at the beginning of mashing to 5.5 ± 0.1. Mashing was performed by placing the bottles in a water bath set at 67 °C. The mashes were incubated for one hour after the temperature inside the mashes reached 66 ± 1 °C (45 min from starting point).

The high deviations visible in the results (Figure 5a,b) are attributed to difficulties in evenly mixing the replicate bottles; however, the results are in agreement with former mashing experiments, with the 20% algae-enriched brew yielding the highest glucose and maltose content amongst the algae-containing wort. Fermentation was completed in five to six days, when, on average, 99% of both the glucose and maltose present on the first day of fermentation was consumed by yeast (Figure 5c,d). After maturation, the beers were analysed in the Anton Paar Alcolyzer (Table 3). 

Despite algal biomass not hindering fermentable sugar release during mashing, the results clearly show an effect on the final product. Statistically significant (*p* < 0.05) differences between treatments were detected for all the parameters presented in Table 3, except for CO_2_ volume. Thus, higher content of algae resulted in stronger colour values, haze, and in a higher caloric content. Additionally, pH values were higher in all algae-enriched beers compared to the control, whilst remaining below pH 5. Further testing is needed to better isolate and evaluate the effect of adjustment with lactic acid and improve the pH management protocols.

The alcohol content was lower in beers supplied with higher algal content. This is in agreement with the real and apparent degrees of fermentation, where the lowest algal content achieved similar values as the control, whereas high algal addition resulted in lower fermentation efficiency. The volume of CO_2_ is also consequently lower in high-algae beer; however, no significant difference was evident for this variable.

Microalgal biomass application in food has proven to be a complex process because of its impacts on technical and sensory properties of the final products. For example, in savoury biscuits, up to 8.3% *Arthrospira platensis* biomass was added to the dough, resulting in an enhanced protein profile of the food; however, at this concentration, the sensory profile was not appealing [31]. Similar results were reported for other species, with the addition of microalgal biomass contents up to 6% in biscuits resulting in improved bioactive profile, although lower biomass additions yielded better sensory scores [13]. In baking goods, the addition of microalgal biomass above 3% had a disruptive effect on dough rheology, with optimal contents often ranging between 1–2% for the different species used [10,32]. In gluten-free bread, it was possible to successfully integrate 4% *T. chui* biomass with a stabilising effect on dough; however, sensory scores were lower when compared to the 1–2% *T. chui* substitutions [12]. 

In our trials, the addition of higher microalgal content was aided firstly by the nature of beer brewing, which happens in suspension, reducing the impact of microalgae on texture. Secondly, some of the unpleasant aromas, which were clearly perceived at the beginning of brewing, were lost throughout the mashing, boiling, and fermentation processes. Finally, a proportion of microalgal biomass sedimented, and was partially removed from the final product during lautering and later bottling, thus possibly reducing the impact on taste and texture. It is also noteworthy that the addition of higher biomass resulted in thicker mashes, with some difficulties in the mixing and major issues in the lautering process, which led to consistent sediment formation. Thus, despite production at a small scale being successful, upscaling of the process is yet to be optimised.

The tasting by brewing experts highlighted how the algae-enriched beers had a characteristic flavour compared to the controls, with increasing sourness upon higher microalgal biomass addition. The taste of algae was almost negligible at 5% addition, giving the beer a slight floral flavour, with hints of umami, while the algae taste was more distinct with increasing amounts. The addition of 12.5% had a clear umami taste with seaweed and marine flavours. Additionally hints of a sweet aroma, resembling canned corn, could be perceived. The 20% beer had the most intense profile, with a stronger seaweed flavour and aroma, marked syrupy notes, and strong umami taste. In the 5 and 12.5% beers, appearance was similar to the control, whereas green hues were visible in the 20% beer. These preliminary remarks on the profile of microalgae-enriched beer must be integrated in future studies, with a thorough sensory evaluation performed by a trained tasting panel.

Based on this trial, a microalgae content ranging between 5–10% could be the most suitable for commercial purposes. However, testing on a larger scale, with specific brewing equipment is necessary to define in more detail, the colour, texture, and flavour in a final product.

## 4. Conclusions

*Tetraselmis chui* biomass was grown under nitrogen-deprived conditions in 250 L reactors achieving the production of biomass with the desired characteristics for brewing. Stepwise trials were performed to assess the effect of microalgae on the mashing process, culminating in the introduction of up to 20% algal biomass as an active ingredient in brewing. Adjustment of pH before mashing proved to be a key step for successful beer brewing with microalgae. In the present study, starch from microalgae was successfully transformed into fermentable sugars during mashing, and further small-scale brewing trials proved the feasibility of microalgae-enriched beer production, yielding a final product that was palatable and had comparatively distinct sensory properties.

Further research is required to fully understand the role of microalgal biomass in brewing and its effects on the final product. The assessment of different recipes and brewing conditions may yield important information on the behaviour and impact of microalgae on the technical properties, allowing one to further optimise and scale-up production. Contextually, a thorough sensory evaluation must be performed, in parallel to an analysis of volatile compounds, to understand in more depth the effects of algal biomass on sensory properties. Finally, the presence and role of bioactive compounds, such as antioxidants, in the final product could be tested to evaluate the possible benefits for the consumers and impacts on shelf-life.

## Figures and Tables

**Figure 1 foods-11-01449-f001:**
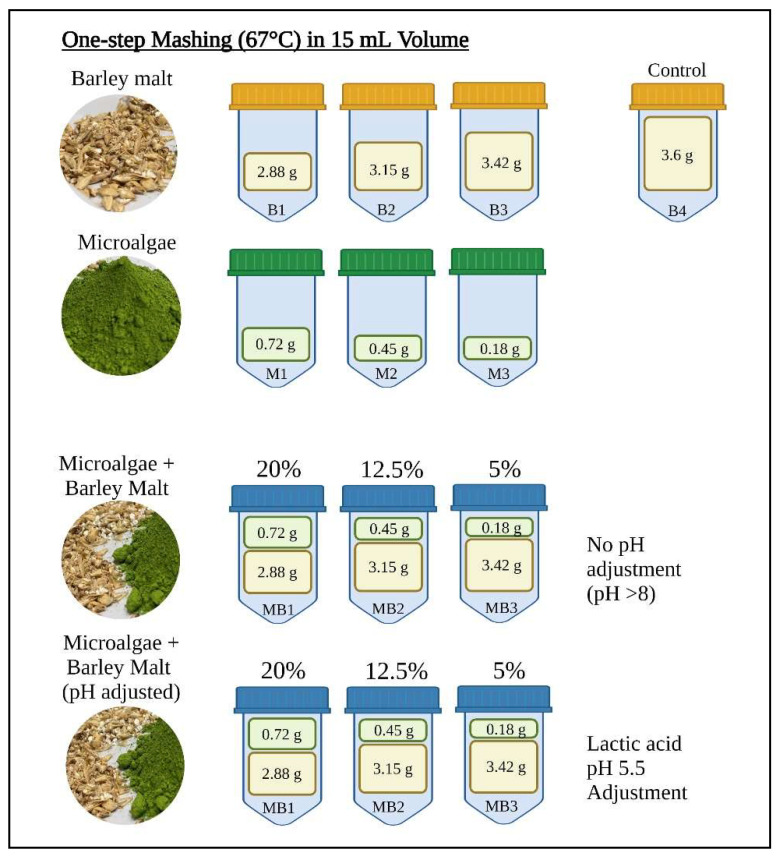
Mashing experiment design. Each tube represents a treatment (performed in triplicate). Yellow tubes (B1–4) represent barley malt mashing trials with content reported in grams; green tubes (M1–3) represent microalgae mashing trials with *T. chui* content reported in grams; blue tubes represent mashing trials with a mixture of barley malt and *T. chui*. Substitution of 20, 12.5, and 5% of the total solids with microalgal biomass, at a solid:liquid ratio of 0.24 kg L^−1^, was performed with (MB1–3) and without (MB1–3) pH adjustment. Created with BioRender.com.

**Figure 2 foods-11-01449-f002:**
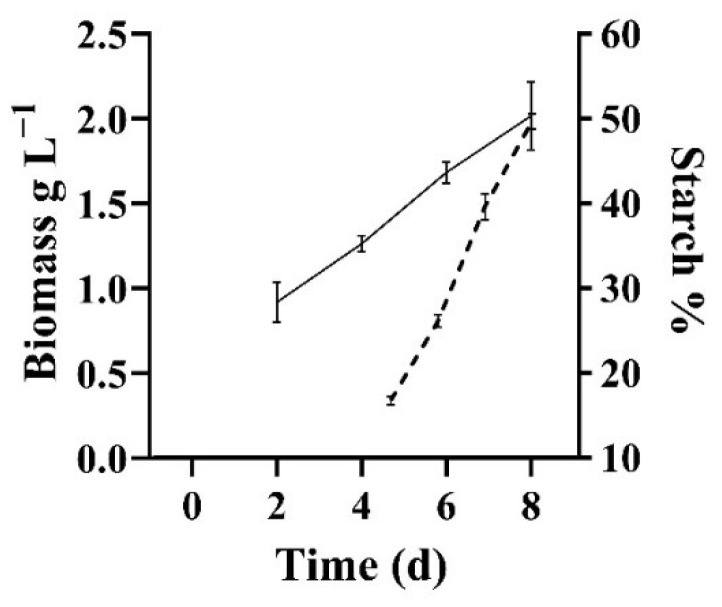
Time course of biomass growth and starch accumulation in *T. chui* cultivated in a 250 L tubular photobioreactor over 8 days. Biomass dry weight is expressed as g L ^−1^ (solid line) and starch accumulation is expressed as % of biomass dry weigh (dotted line). All data represent mean ± SD of triplicate measurements.

**Figure 3 foods-11-01449-f003:**
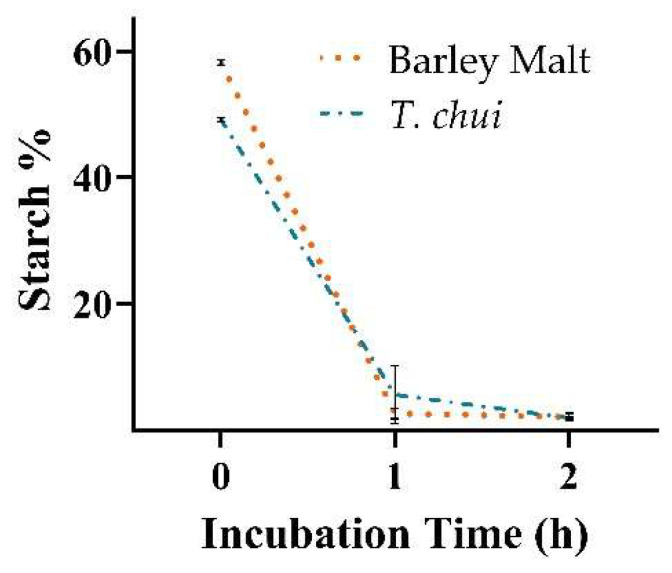
Starch degradation in barley malt (dotted line) and *T. chui* (dashed line) biomass by enzyme-rich wort at 67 °C incubation over two hours. All data represent mean ± SD of triplicate measurements.

**Figure 4 foods-11-01449-f004:**
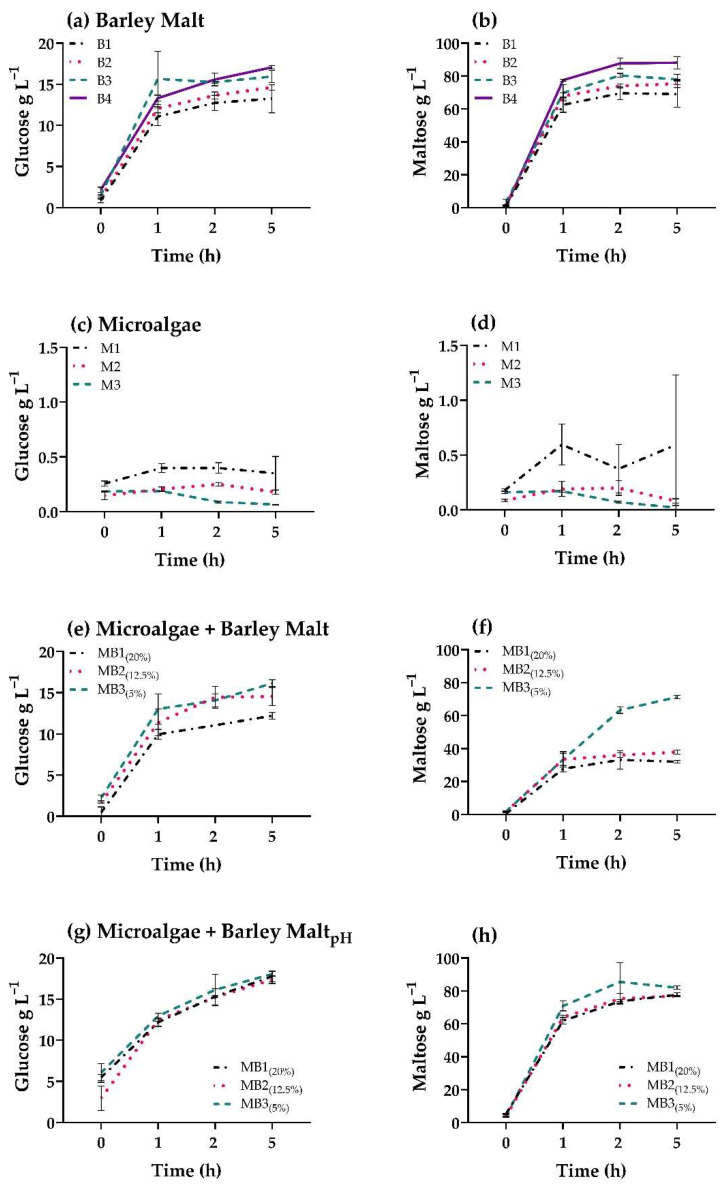
Glucose and maltose accumulation in the wort during mashing experiments with barley malt and starch-rich *T. chui* biomass. (**a**,**b**) Mashing with barley malt; B1-4 correspond, respectively, to 2.88, 3.15, 3.42 and 3.6 g of biomass in 15 mL. (**c**,**d**) Mashing with starch-rich *T. chui* biomass; M1-3 correspond to, respectively, 0.72, 0.45 and 0.18 g of biomass in 15 mL. (**e**,**f**) Mashing with starch-rich *T. chui* biomass and barley malt mixtures without pH adjustment; 20% (MB1), 12.5% (MB2) and 5% (MB3) microalgal biomass was used over 3.6 g of total solids in 15 mL. (**g**,**h**) Mashing with starch-rich *T. chui* biomass and barley malt mixtures with pH adjustment; 20% (MB1), 12.5% (MB2) and 5% (MB3) microalgal biomass was used over 3.6 g of total solids in 15 mL. Samples were incubated over five hours at 67 °C. Detailed information regarding samples is described in Figure 1. All data represent mean ± SD of triplicate measurements.

**Figure 5 foods-11-01449-f005:**
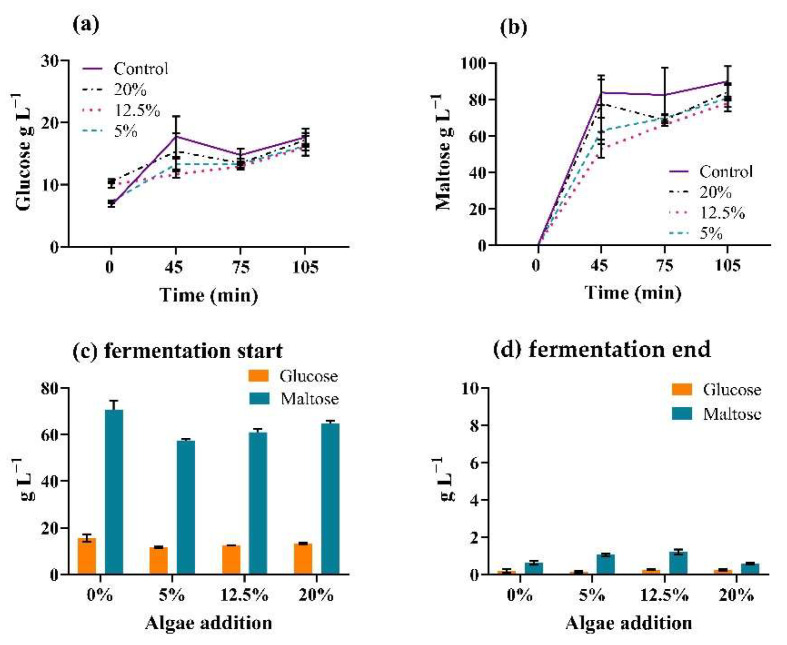
Mashing and fermentation sugar content during brewing trials with *T. chui*. (**a**) Glucose and (**b**) maltose detection during progression of one-step mashing at 67 °C; (**c**) fermentable sugar detection on the first and (**d**) last day of fermentation. All values represent mean ± SD of triplicate samples.

**Table 1 foods-11-01449-t001:** Brewing ingredients for 0.8 L volume. Four beers containing different malt:algae ratios were tested in triplicate.

	Beer 1	Beer 2	Beer 3	Beer 4
Malt %	100	95	87.5	80
Algae %	0	5	12.5	20
Pilsner Malt (g)	182.4	173.28	159.6	145.92
Cara malt (g)	9.6	9.12	8.4	7.68
*Tetraselmis chui* (g)	0	9.6	24	38.4
Northern Brewer 60 min (g)	0.4	0.4	0.4	0.4
Cascade 0 min (g)	1.6	1.6	1.6	1.6
Saaz 0 min (g)	1.6	1.6	1.6	1.6

**Table 2 foods-11-01449-t002:** Composition of starch-rich *T. chui* biomass.

	% Dry Weight	SD
Starch	49.42	0.82
Proteins	22.41	1.57
Fatty Acids	5.20	0.56
SFA	1.13	
MUFA	1.35	
PUFA	2.71	

**Table 3 foods-11-01449-t003:** Anton Paar Alcolyzer results from brewing trials, presented as mean ± SD, using three different algae:malt mixtures vs. control.

Microalgae Content	Alcohol	Haze	Er	Ea	Colour	Calories	pH	CO_2_
	%*v*/*v*	EBC	%*w*/*w*	%*w*/*w*	EBC	kJ 100 mL^−^^1^	-	vol
Control, 0%	5.62 ± 0.14	4.1 ± 4.07	4.13 ± 0.04	2.1 ± 0.03	7.55 ± 1.36	191.09 ± 3.71	4.15 ± 0.02	1.88 ± 0.31
5%	5.61 ± 0.06	3.23 ± 0.17	4.41 ± 0.13	2.38 ± 0.12	8.22 ± 0.11	195.08 ± 3.2	4.4 ± 0	1.88 ± 0.08
12.5%	5.46 ± 0.09	3.47 ± 1.63	5.04 ± 0.09	3.09 ± 0.13	9.3 ± 0.36	201.41 ± 0.56	4.6 ± 0.02	1.8 ± 0.21
20%	5.11 ± 0.07	4.38 ± 0.24	5.45 ± 0.06	3.62 ± 0.04	9.69 ± 0.81	199.83 ± 2.26	4.87 ± 0.03	1.73 ± 0.18

EBC: European Brewing Convention standard unit; Er: real extract; Ea: apparent extract; vol: CO_2_ volume.

## Data Availability

The data presented in this study are available on request from the corresponding author.

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
