# Peer review of "Starch-Rich Microalgae as an Active Ingredient in Beer Brewing"

_foods, 2022, doi:10.3390/foods11101449_

Round 1
Reviewer 1 Report
I would like to thank the authors for their efforts in performing such an interesting study.
Some remarks are given below, and other minor suggestions are given:
Abstract - needs to be re-written not clear representation of the aim of the paper.
Introduction - I would suggest to add a single paragraph in the introduction section about the available literature dealing with this topic.
Materials and methods - are well structured and comprehensive.
The results and the discussion are well expressed.
Conclusion - are not adequately supported by the results obtained. Please improve this section. I recommend the authors put de something adds two sentences referring them to future studies.
Author Response
The authors would like to thank the reviewer for their positive and constructive remarks about the manuscript and for their useful suggestions.
The abstract was re-written to better match the aims of the article.
Concerning the introduction, we agree that the topic of microalgae application in beer brewing was not adequately covered in the manuscript. To the best of our knowledge there is no specific literature about brewing with microalgae, however the discussion of the few available sources exploring microalgal addition to alcoholic beverages has been expanded, to better frame our work.
We improved the conclusions, being careful our statements adequately match the results achieved in our studies and we expanded the discussion about future studies.
Reviewer 2 Report
The manuscript under review is an interesting work on using microalgae rich in carbohydrates as a fraction of the feedstock for the production of beer. The work is novel and the results shown relevant to the Foods journal. In general the quality of the work is very good. There are only some minor comments:
Figure 2a could be deleted since in Figure 2b biomass and carbohydrates production patterns are shown.
In table 3 there would be nice if authors could provide the +- standard deviation to facilitate the readers regarding the consistency of the replicates. If would be better if the results could be tested for statistical significant effects. The latter comment is not mandatory to be addressed, however a statistical testing could have some merit.
Author Response
We would like to thank reviewer 2 for their encouraging words and positive comments about our research study. We agree that figure 2a may be redundant thus we have provided to remove it.
We have added standard deviations to Table 3 and tested the results for significance through analysis of variance.
Reviewer 3 Report
this manuscript describes brewing characteristics of starch-rich microalgae Tetraselmis chui. they have established the starch-accumulating conditions and investigated the brewing profile using malt.
on the whole the ms is well described and performed, but has several concerns
there is not fig5
sensory profiles should be described as a table
Author Response
We’d like to thank reviewer 3 for their effort in reviewing our work. Figure 5 is, to our knowledge, present in the manuscript available on the journal’s webpage. Figure 5 is also correctly cited in the text. Thus, we think Reviewer 3 may have encountered some issues upon downloading the document.
For what concerns the description of sensory profiles we agree it would be clearer to express results in a table, however we think this may be misleading to the readers. The information about sensory characteristics presented in this work are the result of a small tasting by few brewing experts. The results are thus provided as general remarks since there has not been a through sensory evaluation by a trained tasting panel producing replicable and statistically relevant data. We have however addressed the concerns of reviewer 3 by further clarifying the nature of the sensory information we have collected.
Reviewer 4 Report
-This study has novelty and it can play an important role in the beer industry in the future.
-The experiments are planned well and the results and discussion give a clear picture of the study.
-I am satisfied with the language and how the paper has been written and it can be accepted in its present form.
-Only one suggestion is a graphical abstract can be added, which will help the readers.
Author Response
We thank the reviewer for their supportive comments and thorough assessment of our manuscript. We agree a graphical abstract may be beneficial to the readers and have added one in this version of the manuscript.